

# Determining the plant-pollinator network in a culturally significant food and medicine garden in the Great Lakes region

Shelby D. Gibson[1], Thomas M. Onuferko[2,3], Lisa Myers[4] and Sheila R. Colla[4]

[1] Department of Biology, York University, Toronto, Ontario, Canada
[2] Department of Biological Sciences, University of Toronto, Scarborough, Toronto, Ontario, Canada
[3] Canadian Museum of Nature, Ottawa, Ontario, Canada
[4] Faculty of Environmental and Urban Change, York University, Toronto, Ontario, Canada

Corresponding author
Shelby D. Gibson,
shelbydgibson@gmail.com

## ABSTRACT

Understanding the interactions between plants and pollinators within a system can provide information about pollination requirements and the degree to which species contribute to floral reproductive success. Past research has focused largely on interactions within monocultured agricultural systems and only somewhat on wild pollination networks. This study focuses on the culturally significant Three Sisters Garden, which has been grown and tended by many Indigenous peoples for generations in the Great Lakes Region. Here, the plant-pollinator network of the traditional Three Sisters Garden with the inclusion of some additional culturally significant plants was mapped. Important visitors in this system included the common eastern bumble bee, *Bombus impatiens* Cresson (Hymenoptera: Apidae), and the hoary squash bee, *Xenoglossa pruinosa* (Say) (Hymenoptera: Apidae), as determined by their abundances and pollinator service index (PSI) values. Understanding the key pollinators in the Three Sisters Garden links biological diversity to cultural diversity through the pollination of culturally significant plants. Further, this information could be of use in supporting Indigenous food sovereignty by providing knowledge about which wild pollinators could be supported to increase fruit and seed set within the Three Sisters Garden. Our findings can also lead to more effective conservation of important wild pollinator species.

## INTRODUCTION

Pollination is a mutualistic interaction between two levels of the food web—plants and their pollinators (*Jordano, 1987*; *Carvalheiro, Barbosa & Memmott, 2008*; *Ings et al., 2009*; *Willis Chan & Raine, 2023*). Network theory has been used in the evaluation of mutualistic interactions, and the interactions are cumulatively referred to as a plant-pollinator network (*Jordano, 1987*; *Ings et al., 2009*; *Kaiser-Bunbury & Blüthgen, 2015*; *Jolls et al., 2019*). While plant-pollinator network characteristics (such as asymmetry and nestedness) (*Bascompte et al., 2003*; *Bascompte, Jordano & Olesen, 2006*; *Montoya, Pimm & Solé, 2006*) make them

theoretically robust, there is potential for anthropogenically driven environmental disturbance to eventually collapse plant-pollinator networks (*Kearns, Inouye & Waser, 1998*; *Potts et al., 2010*; *Brosi & Briggs, 2013*; *Tucker & Rehan, 2016*; *Tscharntke, 2021*).

Plant-pollinator networks are negatively impacted by various factors including habitat fragmentation and land use changes (*Spiesman & Inouye, 2013*), conventional agricultural practices (*e.g.*, pesticides), non-native species introductions (*Kearns & Inouye, 1993*), and increasingly, climate change (*Memmott et al., 2007*). Documenting these mutualisms and modeling how they respond to change are integral to the conservation and restoration of ecological networks (*Memmott, 2009*; *Kaiser-Bunbury & Blüthgen, 2015*; *Tucker & Rehan, 2016*). Studying plant-pollinator networks helps to fill in baseline information about the ecological role of wild bees and to understand the stability and/or resiliency of the network to environmental change (*Tucker & Rehan, 2016*). The loss of even a single species can have significant effects on reproductive success of the plants within a system (*Brosi & Briggs, 2013*). Declines of pollinators within a network can contribute to negative feedback of less floral reproduction, which then in turn contributes to fewer resources for pollinators (*Tscharntke, 2021*). Wild pollinators provide significant levels of pollination services to crops (*Garibaldi et al., 2011*), yet the details (*e.g.*, the level of pollinator abundance or diversity required to provide adequate pollination) of these relationships remain relatively unknown (*Kovács-Hostyánszki et al., 2017*; *Danforth, Minckley & Neff, 2019*). Pollination deficits are a threat to global food security (*Tscharntke, 2021*).

There have been efforts recently to increase food production without increasing the level of environmental harm from agriculture (*Pretty & Bharucha, 2014*; *Tscharntke, 2021*; *Ramirez & Wright, 2023*). Intercropping is a practice that may increase yield and promote sustainable land and resource use (*Tscharntke, 2021*; *Ramirez & Wright, 2023*). Intercropping has also been suggested as a method of reducing agricultural causes of pollinator decline (*Kovács-Hostyánszki et al., 2017*; *Tscharntke, 2021*). The Three Sisters method of cultivation is a polyculture practice (intercropping) involving the growth of multiple crops simultaneously (*Eames-Sheavly, 1993*; *Kuepper, Dodson & Duncan, 2016*).

Archaeobotanical remnants in the forests and prairies of Canada show evidence of corn domestication as early as 500 A.D. (*Boyd & Surette, 2010*) and common bean and squash cultivation in the Woodland period (1000 B.C.–1000 A.D.) (*Boyd et al., 2014*). The Three Sisters were grown for 500 years pre-contact by the Seneca people in western New York and were referred to as "Diohe'ko", which translates to "these sustain us" (*Lewandowski, 1987*). The Haudenosaunee people (people of the long house) of the Eastern United States and Canada have traditionally planted the Three Sisters Garden (*Eames-Sheavly, 1993*). Broadly, it has been reported that the Three Sisters were grown by all tribes who practiced agriculture in northeastern North America (*Lewandowski, 1987*). This study aims to better understand the pollinator community and plant-pollinator network in a Three Sisters Garden (TSG). The Three Sisters Garden is composed of corn (*Zea mays* L. (Poales: Poaceae)), common bean (*Phaseolus vulgaris* L. (Fabales: Fabaceae)), and squash (*Cucurbita* L. sp. (Cucurbitales: Cucurbitaceae)) (*Boyd et al., 2014*).

*Cucurbita* plants (pumpkins, squash, gourds) are monoecious and rely on insect pollination; each plant has both pistillate (female) and staminate (male) flowers

(*Stapleton, Wien & Morse, 2000*; *Whitaker & Davis, 2012*; *Brochu, Fleischer & Lopez-Uribe, 2021*, *Willis Chan & Raine, 2021a*). Flowers open at dawn and close by noon each day, and pollination must occur within this window (*Nepi & Pacini, 1993*, *Willis Chan & Raine, 2021a*). *Xenoglossa pruinosa* (Say) (Hymenoptera: Apidae) is an oligolectic bee species, foraging only on the flowers of *Cucurbita* crops and wild *Cucurbita* spp. where they are present (*Hurd & Linsley, 1964*; *Willis Chan, 2020*; *Brochu, Fleischer & Lopez-Uribe, 2021*). The hoary squash bee's natural geographic range has increased over the past 1,000 years following the spread of squash planting for agricultural purposes (*Brochu, Fleischer & Lopez-Uribe, 2021*). Domesticated squash has been receiving pollination by wild pollinators prior to the introduction of the western honey bee, *Apis mellifera* L. (Hymenoptera: Apidae) (*López-Uribe et al., 2016*). *Phaseolus vulgaris* is self-compatible, *i.e.*, able to self-pollinate as the flower opens and provides little, if any, nectar (*Ibarra-Perez et al., 1999*; *de Souza Paulino et al., 2023*). It is also noted, however, that the reproductive success of the plant (seed yield) can be increased by visits from larger bees (*e.g.*, carpenter bees, bumble bees, *etc.*) (*Ibarra-Perez et al., 1999*). *Phaseolus coccineus* L. has been found to set few pods without the presence of insect visitors (*Darwin, 1876*; *Free, 1966*; *Free & Racey, 1968*; *Kendall & Smith, 1976*). *Zea mays* is wind pollinated and therefore does not rely on insects for pollination; however, insects may visit the flowers (*Johnson & Hayes, 1932*; *Wheelock, Rey & O'Neal, 2016*; *Rondeau, Willis Chan & Pindar, 2022*).

In some cases, sunflowers would be grown along one side of the Three Sisters Garden (*Kuepper, Dodson & Duncan, 2016*); it has been reported that this was done to attract pollinators to the garden (*Native Seeds Search, 2024a*; *Rodale Institute, 2020*). Other plants, including Hopi tobacco (*Nicotiana rustica* L. (Solanales: Solanaceae)), purple coneflower (*Echinacea purpurea* (L.) Moench (Asterales: Asteraceae)), common milkweed (*Asclepias syriaca* L. (Gentianales: Apocynaceae)), wild bergamot (*Monarda fistulosa* L. (Lamiales: Lamiaceae)), Oswego tea/bee balm (*Monarda didyma* L.), and American vervain (*Verbena hastata* L. (Lamiales: Verbenaceae)), are also planted in some food and medicine gardens (*Our Sustenance, 2020*; *Peel Aboriginal Network and Toronto and Region Conservation Authority (PAN) and (TRCA), 2020*).

The Three Sisters Garden (TSG) is a growing method with long biological and cultural roots, and medicine plants are important to many Indigenous cultures (*Densmore, 1928*; *Lewington, 1990*; *Padulosi, Leaman & Quek, 2004*; *Genuisz, 2015*; *Peel Aboriginal Network and Toronto and Region Conservation Authority (PAN) and (TRCA), 2020*). A better understanding of the pollinator community and plant-pollinator network in the Three Sisters Garden will provide information about the wildlife that provides ecological services to this kind of garden and thus the pollinators that are connected to Indigenous food and medicine sovereignty. Intercropping also offers a sustainable agricultural practice that may be useful, specifically in urban agriculture (*Ramirez & Wright, 2023*).

The objective of this study is to map the plant-pollinator network in the culturally significant Three Sisters garden and determine if and how the pollinator community in the garden differs from the local wild pollinator community based on pan-trap sampling in adjacent, natural sites and in the context of other regional studies. These baseline

conditions will be useful for predicting how the Three Sisters Garden system may be impacted by environmental change into the future.

# MATERIALS AND METHODS

## Study sites and land acknowledgment

This research was undertaken on the traditional territories of multiple First Nations. The campus of York University is located on the traditional territory of the Anishinabe Nation, the Haudenosaunee Confederacy, the Huron-Wendat, and the Métis. The current treaty holders in this location are the Mississaugas of the New Credit First Nation, and the land is subject to the Dish with One Spoon Wampum Belt Covenant. The eastern Ontario field sites are located on unceded Algonquin territory and are subject to Treaty 27 and Treaty 27 1/4. We acknowledge the generations of caretaking of the land by the many Indigenous peoples who have and still do call these places home. Plot A is located in Arnprior, Ontario, Plot B is located in Pakenham, Ontario, and Plot C is located in Lanark, Ontario. A map of sampling sites (Fig. 1) was generated in R using the packages raster (*Hijmans, 2022*), rgdal (*Bivand, Keitt & Rowlingson, 2022*), and sf (*Pebesma, 2018*), with the spatial data for political boundaries downloaded from GADM data (https://gadm.org/data.html) and lakes downloaded from Natural Earth (https://www.naturalearthdata.com/).

## Planting

Seeds for this study were sourced from Urban Harvest (www.urbanharvest.ca) and included scarlet runner (*Phaseolus coccineus*) and true red cranberry beans (*Phaseolus vulgaris*), delicata (*Cucurbita pepo* L. 'Delicata') and pattypan squash (*Cucurbita pepo* 'Patty Pan'), bloody butcher corn (*Zea mays* 'Bloody Butcher'), purple coneflower (*Echinacea purpurea*), bee balm (*Monarda didyma*), and sunflowers (*Helianthus annuus* L. 'Autumn Beauty' (Asterales: Asteraceae)). Pots with purple coneflower, Oswego tea, and American vervain (*Verbena hastata*) were added to the garden sites since growing these plants from seeds is a multi-year process. Sunflowers and medicine plants were added to replicate the addition of these plants to the traditional TSG system. Seeds (including corn, squash, and beans) were started indoors in April using Organic ProMix and Jiffy pots (4″). The soil was kept moist until germination, and light was provided using a TOLYS 1,000 W LED light. At each site, the earth was tilled with a shovel, and soil amendments were added as follows: two 85 L bags of Organic ProMix and four bags of composted sheep manure per plot.

Traditional planting methods were followed (*Native Seed Search, 2024b*), which included mounds of soil placed with the centre of each mound 5 ft from the centre of the next mound for a total of 18 mounds. Each mound was 18″ across and 12″ high. Four corn plants were planted in a 6″ square in alternating mounds. Beans were planted 3″ from each corn plant, creating a square in each corn/bean mound. Squash plants were planted in the remaining mounds, with two plants spaced 4″ apart (alternating mounds of 'Patty Pan' and 'Delicata' squash). Plots A, B, and C in eastern Ontario were used, and each contained a 30 ft × 30 ft garden plot of the Three Sisters. Various medicine plants were also added to all

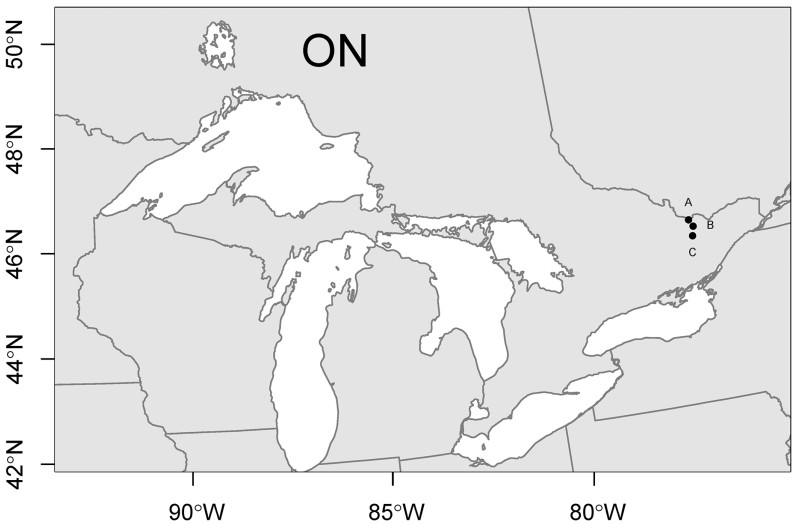

**Figure 1** **Map of study sites.** Location of study sites (A–C) in the great lakes region. Map credit: Natural Earth.

three plots, where some grew more succesfully than others. Plot A also had purple coneflower, Oswego tea, milkweed, and American vervain growing in the garden. Plots B and C also had purple coneflower, bee balm, and sunflowers.

## Insect interaction observations, collection and curation

In 2020, for 4 days per week and over 8 weeks, sweep-net sampling for the plant-pollinator network was conducted, rotating between morning and afternoon sessions, resulting in a total of four sampling sessions per garden per week at three sites in eastern Ontario (Plots A, B, and C) (total of 96 sampling sessions). Morning sessions were between 8 AM and noon, and afternoon sessions between 1 PM and 5 PM (as in *Tucker & Rehan, 2016*). Previous work sampling the plant-pollinator network in a garden setting used similar methods (*Gotlieb, Hollender & Mandelik, 2011*). The garden plot was sampled by walking down each of the rows and up the next row for 20 min. Each pollinator on a flower or inflorescence was collected using a sweep net and placed in an insect vial, and the plant species was noted on the vial. It was not determined whether pollen was deposited; rather, a visit to the flower was used to signal an interaction. Vials were placed in a small lunch cooler containing icepacks until the end of the 20-min session. At the end of each session, the cooler was emptied, with each specimen that could be identified on the wing identified and released (Apis mellifera and Bombus Latrielle spp. (Hymenoptera: Apidae)). A voucher of each species was prepared to represent each species collected in the field (*Kearns & Inouye, 1993*; *Packer et al., 2018*). Each specimen not identified in the field was labelled with the floral species, site, and date. Collected specimens were placed in 70% ethanol (*Kearns & Inouye, 1993*) and pinned at the end of each sampling day throughout the season.

Pan trap sampling was conducted in the natural areas adjacent to the gardens—Pan traps were placed 500 m away from the garden site at two sites. Small plastic coloured

bowls (yellow, blue, and white) were placed on the surface of the ground and filled with soapy water. A 100 m transect was used, and pan traps were placed every ~15 m in a repeating order of yellow, blue, and white (for a total of six traps per transect). Pan traps were set in place for 24-h periods on sweep sampling days.

## Bee identification

Vouchered collected specimens are stored at Dr. Sheila Colla's laboratory at York University, Toronto, Canada. The records were identified to species or morphospecies and include preserved, physical, and some imaged specimens (*e.g.*, *Xenoglossa* Smith and *Bombus*). Pan trap samples were identified to species. Bees were identified to genus (and species for genera that are monotypic in Eastern Canada) using the key of *Packer, Genaro & Sheffield (2007)*. Species-level identifications were made with reference to the keys and taxon concepts of *De Silva (2012)* for *Coelioxys* Latreille (Hymenoptera: Megachilidae); *Gardner & Gibbs (2022)* for metallic weak-veined *Lasioglossum* Curtis (Hymenoptera: Halictidae); *Gibbs et al. (2013)* for non-metallic weak-veined *Lasioglossum*; *Laverty & Harder (1988)* and *Williams et al. (2014)* for *Bombus*; *McGinley (1986)* for strong-veined *Lasioglossum*; *Mitchell (1960)* for *Halictus* Latreille (Hymenoptera: Halictidae); *Mitchell (1962)* for *Melissodes* Latreille (Hymenoptera: Apidae) and *Osmia* Panzer (Hymenoptera: Halictidae); *Onuferko (2017, 2018)* for *Epeolus* Latreille (Hymenoptera: Apidae); *Oram (2018)* for *Hylaeus* Fabricius (Hymenoptera: Colletidae); *Portman et al. (2022)* for *Augochlora* Smith (Hymenoptera: Halictidae) and *Augochlorella* Sandhouse (Hymenoptera: Halictidae); *Rehan & Sheffield (2011)* for *Ceratina* Latreille (Hymenoptera: Apidae); *Mitchell (1962)* and *Rowe (2017)* for non-*Osmia* Osmiini; *Mitchell (1962)* and *Sheffield et al. (2011)* for *Megachile* Latreille (Hymenoptera: Megachilidae); and *Stephen (1954)* and *Mitchell (1960)* for *Colletes* Latreille (Hymenoptera: Colletidae). It was not possible to distinguish some females of *Ceratina dupla* Say from *C. mikmaqi* Rehan & Sheffield, so they were treated as a single morphospecies in data analysis, as *Ceratina dupla/mikmaqi*.

## Statistical analysis

Statistical analysis was conducted in R (R version 3.6.2 (2019-12-12) (*R Core Team, 2019*)). The R package 'vegan' (*Oksanen et al., 2020*) was used to run species accumulation estimates from abundance data (*Gardener, 2014*). The R package 'bipartite' (*Dormann, Gruber & Fründ, 2008*) was used to run community (function *networklevel*) and species (function *specieslevel*) level network analyses. Data were pooled together from all three sites. Community level analysis was conducted including calculations of weighted nestedness and weighted connectance (*Tucker & Rehan, 2018*). Nestedness refers to the level of overlap between generalist and specialist interactions, where values closer to "1" indicate a high degree of overlap and values closer to 0 indicate a low degree of overlap (*Tucker & Rehan, 2018*; *Delmas et al., 2019*). Connectance refers to the proportion of possible interactions that have been realized in the network (*Kearns, Inouye & Waser, 1998*; *Tucker & Rehan, 2018*; *Delmas et al., 2019*) and is a measure of the community's ability to respond to change such as species loss (*Dunne, Williams & Martinez, 2002*;

*Tucker & Rehan, 2016*). Species level analysis included estimation and comparisons of weighted degree and Pollination Service Index (PSI) values (*Dormann, Gruber & Fründ, 2008*; *Tucker & Rehan, 2018*). Degree measures the diet breadth of the pollinators or the number of unique interactions of floral visitors (*Tucker & Rehan, 2016*). The PSI measures the relative importance of each pollinator species to the functioning of the community (1 = the species is critical to ecosystem functioning, 0 = community could function without the species) (*Dormann, Gruber & Fründ, 2008*). PSI is calculated by first determining the proportional representation of the plant species visited by a pollinator and second the proportional representation of the bee species that visit a plant (*Dormann & Fründ, 2024*). Third, these proportions are multiplied to determine the PSI value for each pollinator species (*Dormann & Fründ, 2024*).

## RESULTS

### Abundance and diversity

A total of 310 interactions were observed during the sampling period (July–August 2020) across all sites. Thirty-seven bee species/morphospecies were identified at all sites combined (Fig. 2), with 14 recorded at Plot A, 19 at Plot B, and 22 at Plot C. A total of ten plant varieties were present across all sites combined (Fig. 2), with eight species at Plot A, five species at Plot B, and seven species at Plot C. The species accumulation estimate using the Chao 1 and ACE tests was 63 (se.chao1 = 18) for all plots combined. The observed 37 species represent 59% of the ~63 bee species estimated to occur in the plant community. Bees of the following three families were recorded: Apidae, Megachilidae, and Halictidae. No Andrenidae or Colletedae were recorded in the interaction sampling. The most frequent family was Apidae ($n$ = 200), the most frequent genus was *Bombus* ($n$ = 93), and the most frequent species were *Bombus impatiens* ($n$ = 88) and the hoary squash bee, *Xenoglossa pruinosa* ($n$ = 81). The Shannon's Diversity for the garden sampling was 2.37.

A total of 397 bee specimens were collected from pan-trap sampling. Bees of 19 genera and 53 species were collected in the pan trap samples. Bees of five families were collected (Apidae, Megachilidae, Andrenidae, Colletidae, and Halictidae). Halictidae ($n$ = 228) was the most frequent bee family collected and Andrenidae was the least frequent (Table 1). *Halictus* ($n$ = 88) was the most frequent genus, and *Halictus ligatus* ($n$ = 68) was the most frequent species (Table 1). The species accumulation estimate using the Chao 1 and ACE tests was 62 (se.chao1, 6). The observed 53 species represent 85% of the ~62 bee species estimated to occur at the sites. The Shannon's Diversity for the pan trap sampling was 3.22. Fourteen genera were recorded in the blue pan traps, 15 genera in the white pan traps, and 12 genera in the yellow pan traps (Table 2).

### Plant-pollinator network

For the garden sampling, the most common interactions observed across sites were between *Cucurbita pepo* 'Patty Pan' and *X. pruinosa* (53 interactions) and *B. impatiens* (54 interactions) (Fig. 2). Weighted connectance was 0.08 and weighted nestedness was 0.47. *Bombus impatiens* had the highest degree (6) and *X. pruinosa* (42) and *B. impatiens* (41) had the highest PSI (Table 3).

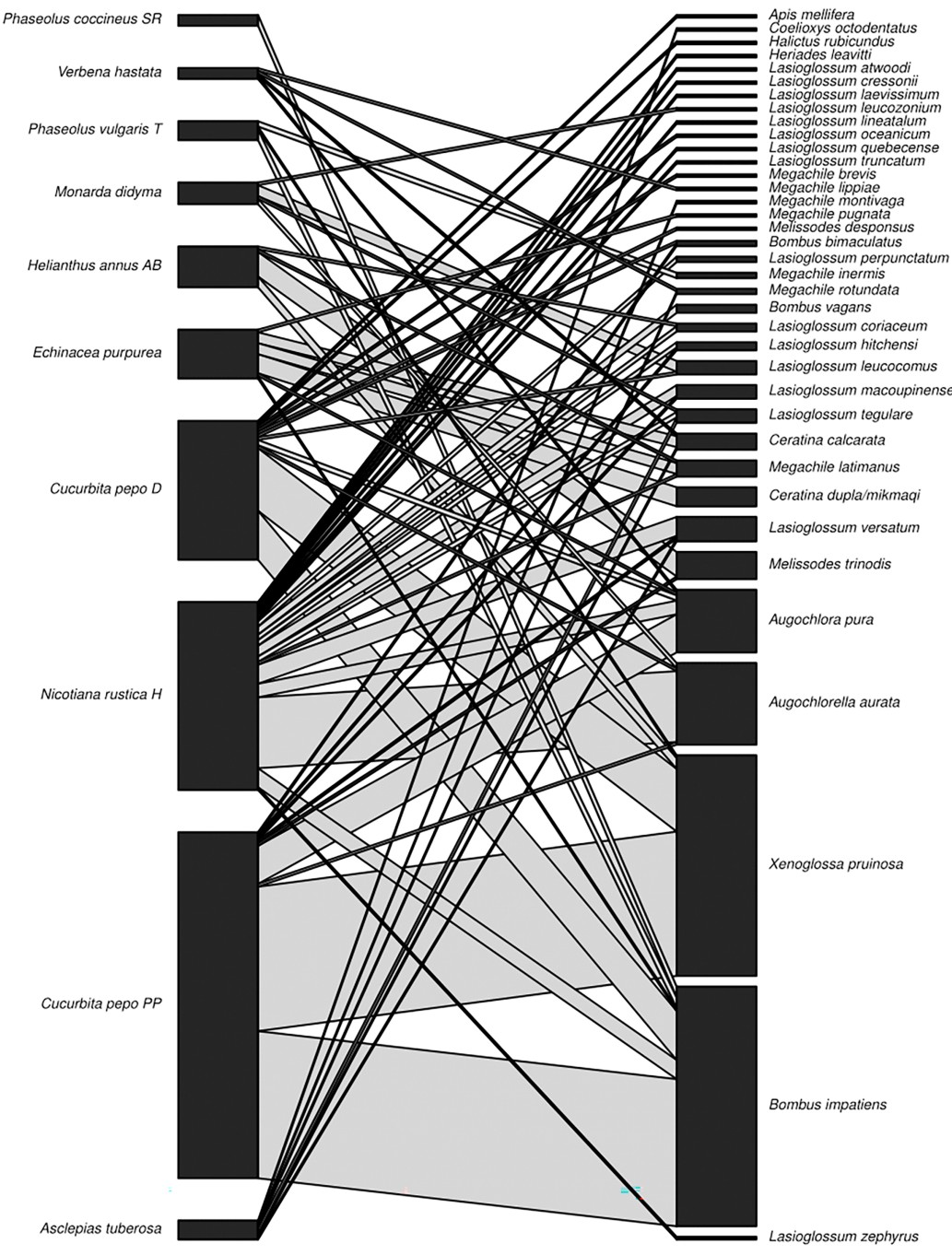

**Figure 2 Plant-pollinator network.** Interaction network displaying plant (species level) and pollinator (species level) interactions (*n* = 310) observed in the three sisters garden plots. Interaction line width is proportional to abundance.                               

*Cucurbita pepo* 'Patty Pan' was associated with the highest number of bees (127), followed by *Nicotiana rustica* 'Hopi', which was associated with the second highest number (69) (Fig. 3). *Phaseolus coccineus* 'Scarlet Runner' and *Verbena hastata* were associated with the lowest number of bees (four) (Fig. 3). *Nicotiana rustica* 'Hopi' was associated with the highest number of bee species (19), followed by *C. pepo* 'Patty Pan' (11)
**Table 1 Bee species in pan traps.** List and abundance of bees collected during pan trap sampling.

| | | |
|---|---|---|
| **Andrenidae** | | **3** |
| | ***Calliopsis* Smith** | **3** |
| | *andreniformis* Smith | 3 |
| **Apidae** | | **103** |
| | ***Apis* L.** | **2** |
| | *mellifera* L. | 2 |
| | ***Bombus* Latreille** | **37** |
| | *borealis* Kirby | 1 |
| | *griseocollis* (De Geer) | 11 |
| | *impatiens* Cresson | 9 |
| | *rufocinctus* Cresson | 8 |
| | *terricola* Kirby | 4 |
| | *vagans* Smith | 4 |
| | ***Ceratina* Latreille** | **24** |
| | *calcarata* Robertson | 10 |
| | *dupla* Say | 1 |
| | *mikmaqi* Rehan & Sheffield | 13 |
| | ***Epeolus* Latreille** | **1** |
| | *scutellaris* Say | 1 |
| | ***Xenoglossa* Smith** | **2** |
| | *pruinosa* (Say) | 2 |
| | ***Melissodes* Latreille** | **37** |
| | *desponsus* Smith | 18 |
| | *druriellus* (Kirby) | 1 |
| | *illatus* Lovell & Cockerell | 3 |
| | *subillatus* LaBerge | 13 |
| | *trinodis* Robertson | 1 |
| **Colletidae** | | **6** |
| | ***Colletes* Latreille** | **3** |
| | *latitarsis* Robertson | 3 |
| | ***Hylaeus* Fabricius** | **3** |
| | *mesillae* (Cockerell) | 1 |
| | *modestus* Say | 2 |
| **Halictidae** | | **228** |
| | ***Augochlora* Smith** | **2** |
| | *pura* (Say) | 2 |
| | ***Augochlorella* Sandhouse** | **62** |
| | *aurata* (Smith) | 62 |
| | ***Halictus* Latreille** | **88** |
| | *confusus* Smith | 16 |
| | *ligatus* Say | 68 |
| | *rubicundus* (Christ) | 4 |

| | | |
|---|---|---:|
| | *Lasioglossum* **Curtis** | **76** |
| | *coriaceum* (Smith) | 4 |
| | *cressonii* (Robertson) | 3 |
| | *hitchensi* Gibbs | 11 |
| | *imitatum* (Smith) | 1 |
| | *leucocomus* (Lovell) | 7 |
| | *leucozonium* (Schrank) | 8 |
| | *lineatulum* (Crawford) | 2 |
| | *oceanicum* (Cockerell) | 7 |
| | *pectorale* (Smith) | 4 |
| | *perpunctatum* (Ellis) | 1 |
| | *pilosum* (Smith) | 2 |
| | *tegulare* (Robertson) | 2 |
| | *versatum* (Robertson) | 20 |
| | *zephyrus* (Smith) | 3 |
| | *zonulus* (Smith) | 1 |
| | *Sphecodes* **Latreille** | **1** |
| | sp. | 1 |
| **Megachilidae** | | **56** |
| | *Coelioxys* **Latreille** | **20** |
| | *rufitarsis* Smith | 20 |
| | *Heriades* **Spinola** | **5** |
| | *carinata* Cresson | 4 |
| | *leavitti* Crawford | 1 |
| | *Hoplitis* **Klug** | **4** |
| | *producta* (Cresson) | 3 |
| | *spoliata* (Provancher) | 1 |
| | *Megachile* **Latreille** | **27** |
| | *brevis* Say | 4 |
| | *campanulae* (Robertson) | 2 |
| | *inermis* Provancher | 2 |
| | *latimanus* Say | 15 |
| | *mendica* Cresson | 1 |
| | *rotundata* (Fabricius) | 3 |
| | *Osmia* **Panzer** | **1** |
| | *distincta* Cresson | 1 |

**Note:**
Bold styling indicates Family and Genus.

(Fig. 3). *Phaseolus coccineus* 'Scarlet Runner' was associated with only one species of bee (Fig. 3). The most common bee species recorded were *B. impatiens* ($n = 88$) and *X. pruinosa* ($n = 81$) (Fig. 4). Seventeen bee species were recorded only once (Fig. 4). There were no interactions recorded for corn flowers.

**Table 2  Bee genera by pan trap colour.** Bee genera collected in blue, white, and yellow pan traps during sampling in the natural environment adjacent to the Three Sisters Garden plots.

**Blue**

*Andrena*

*Augochlorella*

*Augochloropsis*

*Bombus*

*Ceratina*

*Coelioxys*

*Dufourea*

*Epeolus*

*Eucera*

*Halictus*

*Hoplitis*

*Hylaeus*

*Lasioglossum*

*Megachile*

**White**

*Andrena*

*Apis*

*Augochlorella*

*Augochloropsis*

*Bombus*

*Ceratina*

*Coelioxys*

*Dufourea*

*Halictus*

*Hoplitis*

*Lasioglossum*

*Megachile*

*Osmia*

*Protandrena*

*Sphecodes*

**Yellow**

*Andrena*

*Augochlorella*

*Augochloropsis*

*Bombus*

*Ceratina*

*Coelioxys*

*Halictus*

*Hoplitis*

*Hylaeus*

(*Continued*)

|---|
| *Lasioglossum* |
| *Megachile* |
| *Perdita* |

**Table 3 Pollinator service index.** Species level network statistics in the Three Sisters Garden including Degree and Pollinator Service Index (PSI). Degree is the diet breadth of the insect species. PSI is the relative importance of each pollinator species to the functioning of the plant community.

| Insect species | All sites | |
|---|---|---|
| | Degree | PSI |
| *Apis mellifera* L. | 1 | 0.02 |
| *Augochlora pura* (Say) | 5 | 0.1 |
| *Augochlorella aurata* (Smith) | 4 | 0.34 |
| *Bombus bimaculatus* Cresson | 2 | 0.01 |
| *Bombus impatiens* Cresson | 6 | 0.41 |
| *Bombus vagans* Smith | 1 | 0.04 |
| *Ceratina calcarata* Robertson | 3 | 0.21 |
| *Ceratina dupla* Say/*mikmaqi* Rehan & Sheffield | 1 | 0.39 |
| *Coelioxys octodentatus* Say | 1 | 0.14 |
| *Halictus rubicundus* (Christ) | 1 | 0.02 |
| *Heriades leavitti* Crawford | 1 | 0.02 |
| *Lasioglossum atwoodi* Givvs | 1 | 0.04 |
| *Lasioglossum coriaceum* (Smith) | 2 | 0.02 |
| *Lasioglossum cressonii* (Robertson) | 1 | 0.02 |
| *Lasioglossum hitchensi* Gibbs | 2 | 0.02 |
| *Lasioglossum laevissimum* (Smith) | 1 | 0.02 |
| *Lasioglossum leucocomus* (Lovell) | 2 | 0.4 |
| *Lasioglossum leucozonium* (Schrank) | 1 | 0.13 |
| *Lasgioglossum lineatulum* (Crawford) | 1 | 0.02 |
| *Lasioglossum macoupinense* (Robertson) | 2 | 0.05 |
| *Lasioglossum oceanicum* (Cockerell) | 1 | 0.02 |
| *Lasioglossum perpunctatum* (Ellis) | 1 | 0.03 |
| *Lasioglossum quebecense* (Crawford) | 1 | 0.02 |
| *Lasioglossum tegulare* (Robertson) | 3 | 0.15 |
| *Lasioglossum truncatum* (Robertson) | 1 | 0.02 |
| *Lasioglossum versatum* (Robertson) | 3 | 0.1 |
| *Lasioglossum zephyrus* (Smith) | 1 | 0.02 |
| *Megachile brevis* Say | 1 | 0.02 |
| *Megachile inermis* Provancher | 1 | 0.3 |
| *Megachile latimanus* Say | 3 | 0.16 |
| *Megachile lippiae* Cockerell | 1 | 0.25 |
| *Megachile montivaga* Cresson | 1 | 0.14 |

| Insect species | All sites | |
|---|---|---|
| *Megachile pugnata* Say | 1 | 0.06 |
| *Megachile rotundata* (Fabricius) | 2 | 0.2 |
| *Megachile desponsus* Smith | 1 | 0.02 |
| *Melissodes trinodis* Robertson | 2 | 0.5 |
| *Xenoglossa pruinosa* Say | 4 | 0.42 |

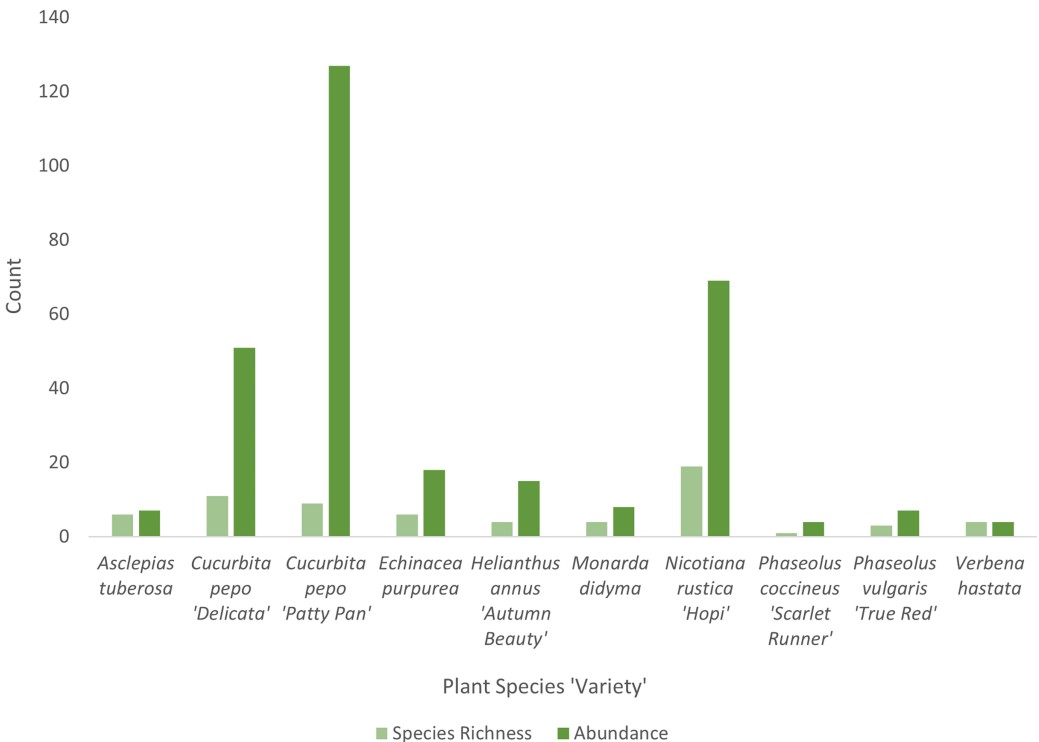

**Figure 3 Bee diversity in garden plots.** Bee species richness and abundance on each plant species in the Three Sisters Garden plots.

## DISCUSSION

The most abundant visitor to the Three Sisters gardens, and the visitor with the highest PSI, was *X. pruinosa*. The most common interactions were between *X. pruinosa* and *C. pepo* 'Patty Pan' and between *B. impatiens* and *C. pepo* 'Patty Pan'. Based on PSI, *X. pruinosa* is one of the key species within the Three Sisters Garden system. There were only two records of *X. pruinosa* in the pan trap samples, which can be explained by this species' specialization on *Cucurbita*. *Xenoglossa pruinosa* had the highest PSI at all sites combined as well as at Plots A and B and was most frequently collected while visiting *C. pepo* 'Patty Pan' and *C. pepo* 'Delicata'.

  As highlighted by *Willis Chan (2020)* and *Willis Chan & Raine (2021a, 2021b, 2023)*, *X. pruinosa* has a close association with *Cucurbita* crops grown for agricultural purposes in

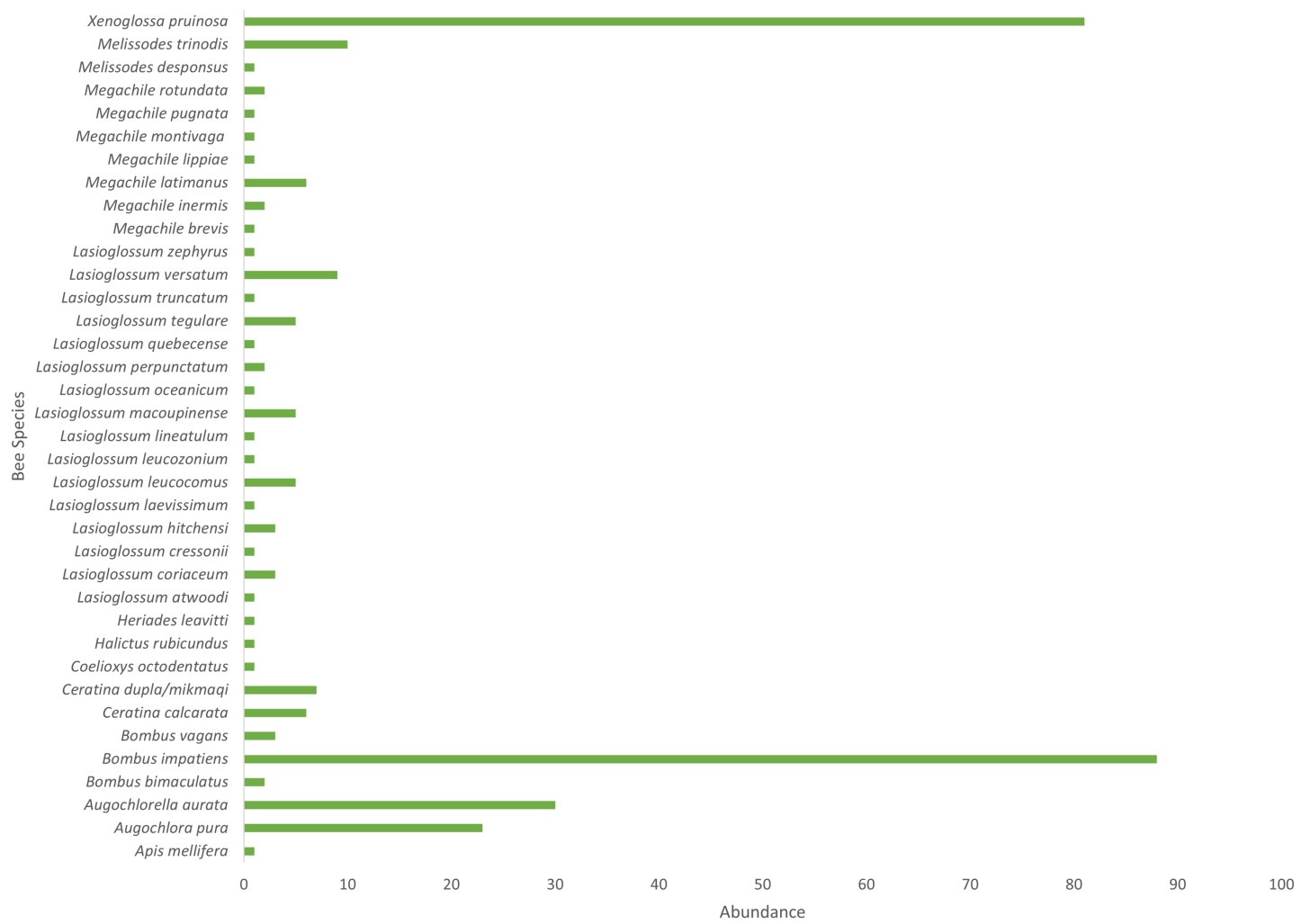

**Figure 4** **Bee abundance in garden plots.** Bee species abundance (*n* = 310) collected in the Three Sisters Garden plots.

Ontario. The range of *X. pruinosa* in Ontario is outside of the range of wild *Cucurbita* (*López-Uribe et al., 2016*). The range of *X. pruinosa* has been found to be impacted by agricultural expansion in North America (*Pope et al., 2023*). The history of the Three Sisters Garden and the remnants of domesticated beans and squash during the woodland period (*Boyd et al., 2014*), as well as the close association still found today between *X. pruinosa* and *Cucurbita* crops, highlight the importance of continued research focusing on managing threats to *X. pruinosa* in a changing environment to ensure continued pollination services by wild bees to this culturally and economically significant food plant. There were also other bee species interacting with *C. pepo* including *Apis mellifera*, *Augochlora pura* (Say), *Augochlorella aurata* (Smith), *Bombus* spp., *Halictus rubicundus* (Christ), *Lasioglossum* spp., *Megachile brevis* Say, and *Melissodes* spp. Future studies should examine the role of these bee species to pollination of *Cucurbita*.

The diversity of bees collected in the natural environment was higher than that in the TSG system. The Shannon's Diversity Index for the garden samples was 2.37, and for the

pan traps was 3.22. Based on the species accumulation estimates, our results represent 59% of bee species estimated to occur in the TSG system, indicating that more extensive sampling is warranted for getting a more accurate representation of the pollinator community. The interaction sampling recorded three bee families whereas pan trap sampling recorded five bee families. Perhaps increased sampling effort would have recorded Andrenidae and Colletidae in the interaction sampling. The absence of *Andrena* Fabricius (Hymenoptera: Andrenidae) (the largest genus of andrenids in Eastern Canada) from our samples may be explained by our sampling having taken place during the summer and *Andrena* being most active in the spring.

Current management recommendations for *X. pruinosa* in agroecosystems are as follows: (1) minimize pesticide exposure, (2) provide nesting sites, (3) maintain yearly field proximity, (4) monitor populations, parasites, and pathogens, and (5) limit deep tillage (*Brochu, Fleischer & Lopez-Uribe, 2021*). Therefore, some of the TSG growing methods may already be implementing some of these practices, such as not using pesticides and limiting deep tillage. With information about the importance of *X. pruinosa* to the TSG and current management recommendations, it is possible to both maintain healthy pollinator populations and ensure adequate pollination of culturally significant food plants. In the absence of squash bees, honey bees and bumble bees are known to be effective pollinators of *Cucurbita* crops (*McGrady, Troyer & Fleischer, 2020*). Bumble bees are also at peak numbers during the pollination window for *Cucurbita* plants (*Willis Chan & Raine, 2021a*). Therefore, using management recommendations for squash bees, such as reducing pesticide exposure (*Willis Chan & Raine, 2021c*), can reduce harm to these important pollinators.

Based on abundance, *B. impatiens* and *X. pruinosa* were the most common floral visitor bee species in the TSG system. *Tucker & Rehan (2016)* found *B. impatiens* to be the most abundant species in their recent study as well. *Bombus impatiens* is a bumble bee species with a broad distribution in North America that is also used as a managed pollinator (including outside its native range) (*Ratti & Colla, 2010*). It has a wide diet breadth, frequently found on both native and introduced plant species (*Williams, Colla & Xie, 2009*; *Colla & Dumesh, 2010*; *Richards et al., 2011*; *Colla et al., 2012*; *Williams et al., 2014*). In this study, *B. impatiens* had the highest overall diet breadth of all bee species, which would explain its ubiquitousness in wild and managed systems. Bumble bees are good pollinators of *Cucubita*, able to deposit more pollen grains per stigma and come in contact with the stigma more frequently than squash bees or honey bees (*Artz & Nault, 2011*). *Willis Chan & Raine (2021a)* found that bumble bees in Ontario are active during the daily crop pollination window, as are the squash bees. While honey bees are found visiting *Cucurbita* flowers, it is likely that the pollen has been depleted and the bees are foraging for nectar (*Percival, 1947*; *Artz & Nault, 2011*; *Brochu et al., 2020*). Pollen supply on staminate flowers has been found to decrease by approximately 60% within the first hour after the flowers open (*Brochu et al., 2020*; *Willis Chan & Raine, 2021a*). This information further highlights the importance of *X. pruinosa* to the TSG system. *Willis Chan & Raine (2021a)* found the pollination window of *Cucurbita* in Ontario to be between 6 AM and 8 AM.

Future studies investigating the role of *X. pruinosa* to the TSG in Ontario should take this into account during experimental design.

Within a pollination network, high connectance and high interaction diversity are associated with stability and resilience (*Tscharntke, 2021*). The overall weighted connectance of 0.08 is the same value found in wild bee pollination networks in northern New England (*Tucker & Rehan, 2016*). The weighted nestedness of 0.47 is similar to this study as well (0.51) (*Tucker & Rehan, 2016*). The authors concluded that both connectance and nestedness were low, indicating that the pollination network in their study may not be resilient to change and may be impacted by significant disturbances (*Tucker & Rehan, 2016*).

Based on the similar results of our study, we deduce that the TSG plant-pollinator network may not be resilient to environmental perturbations. *Brosi & Briggs (2013)* suggest that plant-pollinator networks overestimate the resiliency of pollination networks to perturbations and found that the removal of one pollinator species can affect the quality of reproduction of plant species in the system. *Brosi & Briggs (2013)* suggest that this is because when a pollinator is removed, the plant-pollinator network finds species within the system that will replace the pollination service to the plants visited by this particular pollinator species, but in doing so the network does not consider the pollinator effectiveness of each species. Pollinator effectiveness refers to how well a species of pollinator moves pollen and sets fruits and seeds for a particular plant (*Brosi & Briggs, 2013*; *McGrady, Troyer & Fleischer, 2020*). Therefore, it is important to study not only the plant-pollinator network but also particular interactions between a plant species and its specific pollinators. If *X. pruinosa* were not present, *Bombus* species could provide adequate pollination in terms of pollen grains deposited (*McGrady, Troyer & Fleischer, 2020*), however, and visits would be within the peak pollination window (*Willis Chan, 2020*).

The second most frequently visited plant in this study was *N. rustica*. Hopi tobacco is a culturally significant plant to many Indigenous peoples (*Brokenleg & Tornes, 2013*; *Sadik, 2014*). *Nicotiana rustica* is a plant reported to produce up to two thirds of its seeds through self-pollination and is pollinated by bees (*Mather & Vines, 1952*). In this study, *N. rustica* supported the highest diversity of bee species and the second highest abundance of bee visits. *Augochlorella aurata* represented 16% of the records in the pan trap samples from the wild bee families, suggesting growing *N. rustica* may provide important foraging resources to wild bees. While *N. rustica* is self-compatible (*Mather & Vines, 1952*, *Raguso et al., 2003*), visits by insects increase its reproductive success (*Adler et al., 2012*; *Gibson et al., 2022*). *Augochlorella aurata* belongs to the family Halictidae (sweat bees), which like most other sweat bees, nests in bare soil (*Buckley, Zettel Nalen & Ellis, 2019*), but it is a species that has also been found to be an important pollinator of *N. rustica* (*Gibson et al., 2022*).

Management recommendations for Halictidae includes providing appropriate nesting and foraging resources (*Buckley, Zettel Nalen & Ellis, 2019*). Avoiding tilling the soil is a key factor in providing nesting space for sweat bees (*Buckley, Zettel Nalen & Ellis, 2019*). *Lewandowski (1987)* reported that upon observation of European agriculture, the Seneca

people who had traditionally grown the Three Sisters Garden were shocked by the "wounding of Mother Earth" occurring, which was a reference to the tillage of the soil. In some cases, however, mild tilling has had a positive impact on ground-nesting bee abundance in agricultural areas (*Cusser et al., 2023*).

Beans, forming a large component of the Three Sisters Garden, had some of the lowest observed rates of bee visitation. Only *B. impatiens* was observed visiting *P. coccineus*, despite the other genera being present and visiting other plants within the plant community. Overall, bee visitation to both bean varieties was quite low. Beans are mostly self-pollinated with limited floral resources available, and yield is only marginally increased by insect pollination (*Ibarra-Perez et al., 1999*).

Indigenous and other local communities support pollinator conservation through (1) supporting biocultural (biological and cultural) diversity, (2) landscape management, and (3) diversified farming systems (*Hill et al., 2019*). Here, we examine one of those diversified farming systems. The Three Sisters is a key feature of northeastern North American Indigenous agriculture (*Lewandowski, 1987*). The intercropping growing method of the Three Sisters Garden has been found to support not just humans physically and spiritually but also a wide diversity of wild pollinators through foraging and nesting provisions. Supporting the cultivation of culturally appropriate foods, and therefore food sovereignty, simultaneously supports and depends on the conservation of wild bee species.

## CONCLUSIONS

In this study we aimed to determine the plant-pollinator network in a Three Sisters Garden in the Great Lakes Region. The results of this study highlight *Bombus impatiens* and *Xenoglossa pruinosa* as important pollinators in the TSG system. Three of five bee families found in nearby natural areas were also found in the garden system. One limitation of this study is the lack of sweep net sampling in natural areas, which may be an avenue for future research. Future directions may also include more research efforts focused on the critical role of wild pollinators in culturally significant plants and the applied policies and programs towards promoting their conservation and diversity.

## ACKNOWLEDGEMENTS

We thank Ben Shearer for field assistance on this project, and the private landowners who volunteered their land for the study. Thank you to Dana Prieto, Research Associate for the Finding Flowers Project.

### Funding

This work was supported by the Natural Sciences and Engineering Research Council of Canada (NSERC), (PGSD3-547190-2020) and the Government of Canada's New Frontiers in Research Fund (NFRF), (NFRFE-2018-00485). The funders had no role in study design, data collection and analysis, decision to publish, or preparation of the manuscript.

## Grant Disclosures

The following grant information was disclosed by the authors:

Natural Sciences and Engineering Research Council of Canada (NSERC): PGSD3-547190-2020.

Government of Canada's New Frontiers in Research (NFRF): NFRFE-2018-00485.

## Competing Interests

Dr. Sheila R. Colla is an Academic Editor for PeerJ.

## Author Contributions

- Shelby D. Gibson conceived and designed the experiments, performed the experiments, analyzed the data, prepared figures and/or tables, authored or reviewed drafts of the article, and approved the final draft.
- Thomas M. Onuferko analyzed the data, prepared figures and/or tables, authored or reviewed drafts of the article, and approved the final draft.
- Lisa Myers conceived and designed the experiments, authored or reviewed drafts of the article, and approved the final draft.
- Sheila R. Colla conceived and designed the experiments, analyzed the data, prepared figures and/or tables, authored or reviewed drafts of the article, and approved the final draft.

## Data Availability

The data are available at Mendeley: Gibson, Shelby (2023), "Three Sisters Garden and natural area", Mendeley Data, V1, http://dx.doi.org/10.17632/jr9z897tm7.1.

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
