# Peer review of "Determining the plant-pollinator network in a culturally significant food and medicine garden in the Great Lakes region"

_PeerJ, doi:10.7717/peerj.17401_

## Round 0.1 · original submission · Major Revisions

Once again, thank you for submitting your manuscript to PeerJ and we look forward to receiving your revisions.

I will be happy to accept your paper pending major revisions, detailed by the referees - they are largely focused on clarifying several aspects of your work.

Both reviewers have major reservations and comments about your manuscript. Given this, I would like to see a major revision dealing with all the comments. Please revise paying particular attention to the more critical comments, especially in relation to experimental design and interpretation of the diversity of pollinators or plant visitors.
Please be aware that we consider these revisions to be major, and your revised manuscript will probably have to be re-reviewed.

Revisor 1 basically makes pointed comments within the text and many corrections and changes in the wording of the writing. The most important comments include to a misinterpretation of the data presented in some of the references cited within the text.

Reviewer 2, points out several doubts related to the experimental design and especially to sampling and to the quantification and identity of pollinators and their possible consideration as only visitors.

·

Basic reporting

The paper is well written and much improved. This paper now represents an important contribution to our knowledge of the interactions between bees and Three Sisters Gardens and more specifically to the interaction between bees and beans, tobacco, and Cucurbita in Ontario. Congratulations to the authors. There remain a few suggestions for improvements to cited references (see comments directly on the attached document). Figures are improved--thank you.

Experimental design

The purpose and design of the paper has been clarified. Extraneous confusing information has been removed. Methods have been made more robust and easy to follow. I would like more information about what the natural areas looked like (what plants were there?), how far away the natural areas were from the TSG and what the rationale for that distance is. I still query whether all the sites can be clumped together if the network of plants at each was not the same.

Validity of the findings

The findings of the paper are valid if it is of no consequence that all TSG sites did not have the same plants grown in them. The paper presents important information that is relevant to TSG and may also have relevance to other Cucurbita production methods, especially with respect to the wild bees present in the system that are not Bombus or Xenoglossa. Information about families of bees that are absent from the TSG is also important. There are also interesting findings with respect to tobacco. There are no controls in the data as no experiments were carried out. One of the stated aims of the paper is to compare the network of bees found in TSG to the network of bees in surrounding natural areas. This is an interesting and important idea and the paper's findings show real differences between the two. This implies that bee-plant networks in agroecosystems are different (impoverished?) from those in natural systems.
There is some confusion leading to incorrect conclusions regarding the presence of Bombus impatiens during the pollination window of Cucurbita based on a paper that I authored. I have noted this in comments in the attached document. Perhaps the reference in question should be read more closely.
There is also an understandable drift in focus towards the end of the paper towards recommendations based on monoculture production of Cucurbita. I would suggest that authors note that those recommendations are based on monoculture production and also address how TSG already comply with some of those recommendations.
I have suggested the inclusion of my work on the effects of imidacloprid, a commonly used neonicotinoid pesticide in Cucurbita production as an example of how pesticides negatively impact squash bees in Cucurbita production systems, but I leave the decision as to its inclusion up to the authors.

Additional comments

Thank you for putting in the extra work required to improve your paper. I consider it much clearer and more concise.

Your work is important and in its present form will contribute greatly to our knowledge of the bees associated with TSG as well as our knowledge of wild bee associations with specific plant elements in those gardens (ie tobacco, Cucurbita, beans). I am excited to see information about other wild bees (ie not Bombus, not squash bees) that are associated with Cucurbita as this opens up many more questions about their role in the system. Congratulations.

I have noted a number of small issues directly on the manuscript attached. Please make revisions based on those comments.

Reviewer 2 ·

Basic reporting

No comment

Experimental design

This paper presents an interesting investigation into plant-pollinator networks in the Three-sisters system in Canada, it provides information about which pollinators have higher visiting frequencies and which plants are more visited. However, the experimental design is not very clear, it is not well explained why did they use three plots and why were they sown with different accompanying plants. This lack of explanation can potentially obscure the rationale behind the experimental setup and hinder the interpretation of results. Providing more clarity on these aspects would strengthen the study's methodology section and improve the overall comprehensibility of the research.
Also insect sampling lacks of detailed explanation, did you sample only four days per plot? Or did you repeat this procedure several times during the season?

Validity of the findings

In your result section you report 59% representation of the bee community, this is not very high, therefore you should take your results with caution. This figure is telling you that you should sample more extensively to get a better picture of bee diversity in the garden, please take that into consideration.
Regarding the network analysis, in order to assess the significance of the community parameters you are discussing you should test you results against null models, please see the details below.
Throughout the text you talk about pollinators, however you did not check that pollination was actually occurring, therefore it will be more appropriate to talk about visitors. Of course, if you have a high frequency of visitation the probability of pollination to occur is high but several studies have found that many of these frequent visits do not translate into actual pollination. I suggest you consider this caveat throughout the manuscript and suggest ways of improving.

Additional comments

Minor comments:

L 173-176 How many samplings in total? How many weeks?
L268-286. Did you have an estimation of the number of flowering plants per plot per sampling?
L287. Any years? Was the experiment longer than a year? It is not specified in the methods

L256-62. Did corn, squash and beans planted as seed or as seedlings? Please specify because from the above description is not clear

L262-265. Please explain why did the plots were planted with different additional plant species.

L263-270. Did all the bees found in the gardens were present in the pan traps? Are there some specialist to the gardens?

L294-299. Please specify how many pan traps did you use in each transect

L331. Please specify if you analyzed each plot as a different network or if you pooled the data together from the three

L337. Did you test your nestedness values against null models? This is the correct way to assess if your network is significantly nested or not. I suggest you read the paper by
Dormann CF, Fründ J, Blüthgen N, Gruber B (2009) Indices, graphs and null models: analyzing bipartite ecological networks. Open Ecol J 2:7–24

L363. Please specify if the Chao 1 and ACE are from all plot together? It would be interesting to see if each plot have a different coverage or if the sampling effort was similar between plots

L370. It would be interesting to know if you got similar species considering the different pan trap colors.

L382. Again please state if you are considering the three plots together or what was the analytical procedure

L420. Did you sample several years? It is not stated in the methods.

Figure 1. In order to be able to visualize the nested structure, I suggest you order your species with the following command in R:
sortweb( yournetwork, sort.order="dec")

Instead of table 1, I suggest to include a map so people from all over the world can understand where are the plots situated.

---

## Round 0.2 · Minor Revisions

Again, thank you for submitting your manuscript to PeerJ and for making important revisions to the manuscript. According to the reviewer and myself, your manuscript still has a minor revision pending.

I will be happy to accept your article pending this minor revision.

·

Basic reporting

The basic reporting in this revised manuscript are adequate.

Experimental design

no comment

Validity of the findings

no comment

Additional comments

Lines 337-338
This statement is still incorrect. Bumble bees are at peak numbers during the early morning pollination window for Cucubita crops.

Line 340
This statement is also incorrect with respect to Cucurbita crops. The correct analysis would be as follows:

Reducing pesticide use will reduce harm but no statements can be made about increasing populations as well as pollination based on my work. As a matter of fact, even when populations were reduced by ~90%, pollination was not significantly different from year to year in my three-year study. This is likely due to the large staminate to pistillate flower ratio in the crop.

---

## Round 0.3 · accepted · Accept

After evaluating this revised version, I realize that the minor changes suggested by the reviewer were made, so I am happy to accept this interesting manuscript, which is now ready for publication in PeerJ. Congratulations